# Early Management for Fracture-Related Infection: A Literature Review

**DOI:** 10.3390/healthcare12131306

**Published:** 2024-06-29

**Authors:** Giovanni Vicenti, Claudio Buono, Federica Albano, Teresa Ladogana, Elisa Pesare, Giulia Colasuonno, Anna Claudia Passarelli, Giuseppe Solarino

**Affiliations:** Orthopaedic & Trauma Unit, Department of Traslational Biomedicine and Neuroscience (DiBraiN), School of Medicine, University of Bari Aldo Moro, AOU Consorziale “Policlinico”, 70124 Bari, Italy; dott.gvicenti@gmail.com (G.V.); f.albano34@studenti.uniba.it (F.A.); t.ladogana@studenti.uniba.it (T.L.); e.pesare@studenti.uniba.it (E.P.); g.colasuonno10@studenti.uniba.it (G.C.); a.passarelli2@studenti.uniba.it (A.C.P.); giuseppe.solarino@uniba.it (G.S.)

**Keywords:** fracture-related infection, trauma surgery, DAIR

## Abstract

Fracture-related infections (FRIs), as shown in the literature, represent one of the main complications of trauma surgery. They are a consequence of an implant-related “biofilm” formation and are a challenge for surgeons, microbiologists, and infectious disease specialists. For a correct diagnosis, careful clinical evaluation, to look for signs/symptoms attributable to an infectious condition, and instrumental examinations, to highlight the site of infection, its extent, and its severity, are both essential. Unfortunately, due to the lack of data in the literature, there is no consensus about guidelines on the diagnosis and treatment of FRIs. The purpose of this study is to present an up-to-date concept evaluation of the diagnostic procedures and treatment options available in the management of fracture-related infections.

## 1. Introduction

Fracture-related infections (FRIs), as shown in the literature, represent one of the main complications in trauma surgery [1,2]. The severity of this condition can manifest across a spectrum, ranging from a loss of limb function to amputation [3]. This occurs not only in patients with serious comorbidities but also in otherwise healthy individuals [1]. For these reasons, it is important to focus attention on prevention and early diagnosis [3,4].

An FRI is a consequence of an implant-related “biofilm” formation, and the management of this condition poses a real challenge not only for surgeons but also for microbiologists and infectious disease specialists [1,5,6]. The risk of encountering this complication is higher for immunocompromised patients, obese individuals, and those with low socioeconomic status or inadequate hygienic conditions [1]. Vascular diseases and smoking also elevate the risk because they negatively impact the healing of soft tissues and bones [1]. Nowadays, antibiotic therapy achieves remarkable importance as a useful weapon not only on its own but also in addition to surgical treatments [6]. The management of this type of complication depends on different aspects [2,7]. Among these, the conditions of cutaneous tissues and the healing rate of fractures are decisive [7].

A well-done debridement with the removal of all tissue that does not contribute to wound and fracture healing represents one of the main aspects in the management of FRIs [7]. Scar, necrotic, or ischemic tissues have to be excised in order to be replaced with well-perfused and healthy tissues [7]. Moreover, approximately 40% of FRIs require free flaps or vascularized pedicles to guarantee appropriate tissue coverage [7]. A good tissue coverage is represented by a vascularized, epithelized, and space-filling, soft, and not fibrotic tissue capable of covering bone and metal implants. The result is to provide a secure barrier against bacterial contamination [7].

The removal of the implant can only be performed in case of fracture healing or, in a case when the fracture is not healed, the implant could be replaced, ensuring adequate antibiotic coverage [8]. In order to define treatment algorithms for FRIs, a multidisciplinary collaboration between microbiologists, surgeons, pharmacists, infectious disease physicians, and nursing staff is mandatory. Despite the fact that FRIs can have serious consequences, no specific guidelines have been developed.

We use the CDC guideline (US National Institutes for Health Centers for Disease Control and Prevention) to prevent surgical-site infection (SSI), which distinguishes between superficial, deep, and organ infections. 

The purpose of this study is to present an up-to-date concept evaluation of the diagnostic procedures and treatment options available in the management of fracture-related infections in order to bypass the lack of data in the modern literature.

## 2. Materials and Methods

A narrative literature examination was conducted between January 2005 and December 2023 in the PubMed databases. We applied filters to include only English-language studies.

Three authors collected the literature separately (T.L., F.A., C.B.) and then screened abstracts and full texts.

Each paper was evaluated by one independent investigator (G.V.); cases of disagreement were solved by consensus with a third author with extensive expertise in implant-related infections (G.S.). 

Reviews, systematic reviews, and meta-analyses focusing on infection in the orthopedic field were taken into account, and “fracture-related infection”, “early management”, “review”, and “trauma surgery” keywords with the Boolean operators AND and OR were used. 

We excluded studies centered on in vitro or in vivo animal models, articles without full text, papers that made no mention of fracture-related infection treatment, and low-evidence studies (expert opinion, technical comments, and clinical trials).

Using a Microsoft Excel spreadsheet, some information was gathered following the authors’ protocol, including the study’s design, year of publication, first author, and title. 

## 3. Results

The preliminary search of PubMed databases has shown a total of 283 publications. Initially, 36 duplicates were eliminated. After the analysis of titles and abstracts, a total of 57 articles remained. We read the full text of the remaining 57 articles, and 36 were excluded because they did not meet the inclusion criteria. A total of 21 studies that met the inclusion criteria were selected. 

## 4. Discussion

An FRI represents one of the main and devastating complications in trauma surgery [7,9]. In fact, patients affected by FRIs require aggressive antibiotic therapies and, in the worst case, multiple surgical procedures that could be responsible for long-term hospitalization and higher healthcare costs [9]. The incidence of FRIs is up to 25% and depends on the severity of the injury, and the risk is higher for high-energy trauma [9]. In most cases, FRIs are a consequence of an open fracture. Indeed, the risk increases with a higher degree of exposure, according to the Gustilo classification [10]. The Gustilo classification is also used to guide antibiotic treatment and its duration. Along with these, other factors could be responsible for a higher risk of infection, such as co-morbidities, the fracture site, cigarette smoke, diabetes mellitus, male gender, and polytrauma [9,11].

In the pathogenesis of FRIs, a key role is played by the formation of a biofilm on the surface of the implants [1]. The biofilm creates the perfect environment for microorganisms’ survival at higher antibiotic concentrations [1,12,13]. Therefore, the use of a single IV antibiotic does not always allow us to achieve suitable therapeutic levels; even though the use of local antibiotics is debated, combined use with intravenous antibiotics may be the best choice [1]. The infections usually are caused by exogenous microorganisms during trauma events (with a higher risk of contamination in open fractures) or during surgical procedures, such as the implant of fixation devices [1]. Bacteria can reach the fracture site in different ways, such as direct localization at the time of the injury, through a hematogenous route, or during hospitalization (nosocomial infections) [9]. Staphylococcus aureus (30–42%) and coagulase-negative staphylococci (20–39%) represent the majority of microorganisms responsible for monomicrobial infections. Enterobacter, vancomycin-sensitive Enterococcus, and Pseudomonas are frequently involved in polymicrobial infections [14].

In most cases, the infections are polymicrobial, especially after an open fracture [1]. These microbial communities could contribute to the persistently high failure rate in managing infections following trauma [14].

The different phases of the biofilm formation are crucial for determining the efficacy of AB, and another important aspect in the treatment choice is the entity of bone healing and the involvement of soft tissues [1].

Different studies have been published in the literature regarding the mechanism that bacteria use to cause FRIs; for example, S. aureus creates a niche directly in bone canaliculi [9].

Willenegger and Roth developed a classification system of the FRI based especially on the time of the infection onset: early-onset infections occur within 3 weeks from surgery, while delayed- and late-onset, respectively, occur 3–10 weeks and over 10 weeks after surgery. Another classification described in the literature differentiates the FRI in acute or chronic infections according to a limit of 6 weeks after surgery [15].

The risk of infection after surgery goes from 1–2% for closed fractures to 30% for open ones [1]. 

The literature reported several risk factors that could be responsible for implant contamination: male gender, smoking, anatomical site, and entity of the lesion [3]. It is established that smoking hinders the healing of wounds [16]. Therefore, it is advisable to encourage the patient to quit smoking as early as possible before surgery and to abstain from smoking throughout the entire healing process [16]. Endocrine disorders, including diabetes mellitus, may also impact the surgical outcome adversely [16]. Hemoglobin A1c (HbA1c) serves as a measure to assess glycemic control in diabetes mellitus [16]. The risk of infection tends to rise in patients with diabetes mellitus as perioperative HbA1c levels increase [16]. Additionally, in individuals without diabetes, elevated blood sugar levels (hyperglycemia) are linked to unfavorable clinical outcomes [16]. For orthopedic trauma patients admitted, research indicates that stress-induced hyperglycemia is notably and independently associated with an increased risk of infection [16]. Early soft tissue coverage is crucial to prevent infection with the combination of prophylactic antibiotics in the case of open fractures [3]. The literature reports that a wound should be covered within 72 h and no later than 7 days [4]. The use of a single dose of first-generation cephalosporin as prophylaxis therapy before surgery is mandatory [4]. In case of severe soft tissue damage, prophylactic therapy includes a combination of a cephalosporin with another molecule useful against Gram-negative bacteria [4]. But, due to the lack of data in the literature, no guidelines are available about the duration of prophylactic therapy. However, it should not exceed 72 h [4]. In addition, the exculsive use of IV antibiotics in the case of open fractures could not always be enough, so its use in combination with another local antibiotic may be recommended [4]. Morgenstern et al. carried out a meta-analysis that showed a significant reduction in **i**nfection using additional local antibiotics, such as polymethyl methacrylate (PMMA), for open fractures [1].

In 2018, a prestigious and international group of experts developed a consensus regarding FRI’s definition [17]. It addressed the difference between confirmatory criteria and suggestive criteria. The definition of these criteria depends, firstly, on medical history and clinical exams and, secondly, on surgical exploration and microbiological examination [18]. Clinical signs such as fistula/sinus and purulent drainage represent elements of confirmation, while local/systemic signs, e.g., fever, joint effusion, redness, and elevated serum inflammatory markers, are considered suggestive criteria [18]. In addition to clinical evaluation, microbiological and histopathological investigations are mandatory [18]. Govaert et al. [2] reported that the presence of confirmatory criteria should be considered as pathognomonic signs of infections, and there is a lack of studies in the literature regarding the predictive value of suggestive criteria in the diagnostic pathway [2]. So, the suggestive criteria have to be considered not pathognomonic but should lead the surgeon to go on with further and scrutinized exams [2]. In order to achieve early diagnosis, a multistage analysis should be taken into account, including, on one side, laboratory tests, and on the other, medical imaging and microbiological investigations [2]. The confirmation of infection requires microbiological culture tests from tissue samples or metal implants [16]. It is imperative to obtain samples early in the surgical procedure to prevent contamination [16]. Specifically, five deep tissue samples should be gathered from areas surrounding the fracture and adjacent to the implants [16]. The integration of microbiological and histopathological analyses has been demonstrated to improve diagnostic precision, facilitating the implementation of more specific and targeted antimicrobial therapy [16].

With regard to serum inflammation markers, leukocyte count (LC), C-reactive protein (CRP), and the erythrocyte sedimentation rate (ESR) are the most frequently measured [2]. CRP levels peak on the second day and subsequently revert to the baseline after **three** weeks [2]. Instead, the ESR reaches its peak between days 7 and 11 after surgery and gradually declines until the sixth week [2,19].

In a recent review carried out by Van Den Kieboom et al., CRP displayed a sensitivity ranging between 60.0% and 100% and specificity between 34.3% and 85.7%, representing the most reliable inflammatory marker [19]. In addition, the diagnostic predictive value of each of them (CRP, LC, and ESR) taken individually has limits [18]. 

A pilot study performed by Kumar et al. examined the predictive role of alpha defensin in diagnosing fracture-related infections [20].

Alpha defensin, along with leukocyte esterase, are two tests used in the diagnostic pathway for periprosthetic infections [20]. Alpha defensin is tested using the ELISA method, providing results within a few hours [20]. In contrast, leukocyte esterase is measured with colorimetric test strips, yielding results in just 20 min [20]. According to the aforementioned study, alpha defensin levels are significantly higher in patients with early (<1 month of index surgery) post-operative fracture-related infection [20]. 

Regarding the role of histopathology, in a recent study about the value of quantitative histopathology, Morgenstern et al. postulated that the presence of >5 PMN/HPF (polymorphonuclear neutrophils per high-power field) was always associated with infection with a specificity of 100% and a positive predictive value of 100%, while on the contrary, the absence of PMNs had a very high link with aseptic nonunion with a specificity of 98% and a positive predictive value 98% [18]. The integration of clinical signs, microbiological cultures, and bimodal histopathological analysis (distinguishing between absent NPs and ≥5 PMNs/HPF) enhanced diagnostic accuracy in as many as 96.8% of cases [18]. 

Concerning medical imaging, radiography, computed tomography (CT), magnetic resonance imaging (MRI), three-phase bone scan (BS), fluorodeoxyglucose positron emission tomography (FDG-PET), and white blood cell (WBC) scintigraphy are widely used. XR is the first exam required because it is cheap and easily accessible [2]. In the event that the surgeon needs further details, a CT scan can be performed [5]. However, CT exposes patients to a really high radiation dose and has a low discriminatory capacity for infection investigation with a sensitivity of 47% and specificity of 60% [5]. XR and CT can allow for the searching of signs like implant loosening, bone lysis, nonunion, or sequestration [5]. Morphologic bony and soft tissue changes can be better detected using MRI [5]. The reported sensitivity and specificity of MRI for detecting an FRI ranges between 82% and 100% and between 43% and 60%, respectively [5]. With regard to nuclear imaging, the sensitivity and specificity of WBC scintigraphy + SPECT concerning FRI diagnosis are reported to be 79–100% and 89–97%, respectively [5]. A significant advantage of WBC scintigraphy is that its accuracy is not conditioned by surgery [2].

The goal of adequate treatment in an FRI is the recovery of function, bone and soft tissue healing, and a reduction in chronic infections or complications. The strategy adopted in the vast majority of cases is dual, consisting of antimicrobial therapy and surgical treatment.

The surgical approach depends on whether the bone is healed or not. In case of failure of bone healing, the “DAIR” (debridement, antimicrobial therapy, implant retention) procedure might be taken into consideration [21]. On the contrary, if the bone has healed, another approach could be used, and it consists of debridement and the removal/replacement of the implant combined with antibiotic therapy [21].

For what concerns surgical strategies and, in particular, DAIR, the main purpose is to pursue and sustain stability directly in the site of fracture [22]. However, various crucial factors influence the feasibility of a DAIR procedure, such as soft tissue condition, the ability to perform an appropriate surgical procedure, the experience of the surgeons, the susceptibility of the pathogen involved, and the absence of a patient’s comorbidities that could undermine the success of the surgery [22]. Together, these considerations determine the appropriateness of performing a DAIR procedure [22]. Buij et al. performed a retrospective multicenter cohort study at two level 1 trauma centers. Their work was focused on 141 patients with early FRIs [22]. A treatment protocol for managing patients with early-onset FRIs was implemented in both centers [22]. According to these protocols, the preferred approach for early-onset FRIs with stable fracture fixation was an early debridement, antibiotics administration, and the retention of the implant (a DAIR procedure) [22]. Ensuring sufficient soft tissue coverage was deemed as a crucial aspect of the operative process, while intravenous (IV) empiric antimicrobial therapy commenced immediately after the surgical debridement and tissue sampling for microbiological culturing [22]. Comprehensive control of infection was accomplished in 94%, while 19 patients did not experience adequate bone healing of the fracture site, with the consolidation rate higher in the case of polymicrobial injection [22]. However, in 94% of cases, comprehensive infection control was successfully achieved [22]. 

It is important to remember that the type of synthesis itself can also influence the outcome of the treatment; for example, the intramedullary nailing is associated with a higher risk of failure rate in the case of DAIR because it impedes the debridement of the bone canal compared to synthesis with plates [21]. 

It is well established in the literature that in the case of the FRI, the highest success rate (approximately 90%) is achieved if the DAIR protocol is implemented within three weeks of surgery (in acute/early-onset FRIs) [5]. If the infection shows up after 6 weeks, the DAIR strategy has a succession rate of less than 70%; after 10 weeks, it decreases to under 60%, and this seems to be linked to the inability of the antibiotics to eradicate completely an aged biofilm [5]. So, the age of a biofilm or the infection does not define the duration of the antibiotic therapy but is crucial in the choice between DAIR or the complete removal of the implants [5].

There has always been a great deal of interest in the literature in trying to understand if there is a correlation between the duration of infection and the chances of success/failure of a DAIR procedure [18]. In a recent literature review published by Morgenstern et al., data from 276 patients were collected in order to analyze the correlation between the onset of the infection and the efficacy of the surgical procedure [18]. 

The results of this review showed that in cases of short-term infection (up to 3 weeks) and in cases of a patient’s general condition being good, the DAIR protocol is associated with a high success rate (between 86% and 100%) [18]. Success rates drop between 82 and 89% in case of longer time intervals between fracture fixation and FRI revision surgery (between three and ten weeks) [18]. For late infections with a time interval between fracture fixation and FRI revision surgery of more than ten weeks, success rates drop to 67% [18].

Morgenstern et al., therefore, concluded that in acute or early-onset FRIs with a short duration of infection, the DAIR protocol gives excellent results up to 10 weeks after osteosynthesis, whereas late or delayed FRIs may be associated with a higher risk of recurrence after the DAIR protocol [18]. It is equally agreed that in the proper management of FRIs, not only should the time since the surgical fixation be considered, but the management should be multifactorial, including all the aspects of infection [18]. 

It seems clear that the key to any treatment is a radical debridement, the removal of all necrotic material, biopsy samples for microbiological and histological examinations, and adequate soft tissue management [1]. The reconstruction and the coverage of soft tissues after debridement are important for restoring the barrier responsible for the prevention of fracture contamination [23]. The choice of the correct type of flap (local or free flaps) used in the coverage of soft tissue is now the subject of debate in the literature [23].

Nowadays, in the literature, new studies about the use and efficacy of local antibiotics in combination with surgical treatment have been conducted [24]. These local therapies might be used after an extensive debridement in order to achieve a higher level in the site of infection [24]. The most common local antibiotics used are gentamicin, tobramycin, vancomycin, and clindamycin [24]. After surgery, empirical antimicrobial therapy should be started in case of a high suspicion of infection [24].

The choice of empiric antibiotics depends on patients’ risk factors, i.e., previous therapy, allergies, comorbidities, debridement in the same site, and different types of pathogens [25]. Generally, initial empirical therapy should include a lipopeptide or glycopeptide and an anti-GNB agent, and then it should be adjusted as soon as possible, according to the results of the cultures [25]. The duration of target therapy depends on the surgical procedure chosen. In the case of implant retention, therapy should be administered for 12 weeks, while a 6 weeks therapy should be sufficient in case of implant removal [25].

Nowadays the treatment strategies of FRIs are a readjustment of treatments used for other types of implant infection (i.e., periprosthetic joint infection) because of the lack of specific guidelines [26]. According to the literature, intravenous (IV) therapy should be administered not longer than 2 weeks until the patient’s clinical condition stabilizes and the results of culture tests are obtained [26,27].

In the case of staphylococcal infections, rifampicin turns out to be the most suitable antibiotic and should be started immediately after an adequate debridement [28]. In order to reduce the development of microbial resistance, a combination of rifampicin with molecules from other pharmacological classes is mandatory, such as fluoroquinolone [28]. 

Other drugs associated with rifampicin are cotrimoxazole, minocycline, or fusidic acid, despite the limited evidence in the existing literature [1]. 

The use of intravenous (IV) benzylpenicillin is the first treatment choice in case of streptococcal infections, while ampicillin is effective against enterococcal infections [1]. After 1–2 weeks of IV therapy, oral therapy with amoxicillin should be administered [1].

In the case of Pseudomonas infection, the best choice is represented by β-lactam antimicrobials such as piperacillin/tazobactam, cefepime, ceftazidime, or carbapenem [1]. As described in the literature, another important choice for this type of bacteria is levofloxacin. This antibiotic, as showed in the literature, has demonstrated an ability to prevent the growth of resistant bacteria better than other antibiotics in the same class, such as ofloxacin or ciprofloxacin [20]. Despite this, ciprofloxacin represents the only quinolone that can be used as oral therapy against Pseudomonas [20].

DIAR is not the only strategy we have in order to treat an FRI. Another surgical procedure described in the literature is implant removal. 

In their study, Alec S. Kellish et al. [8] sought to determine trends and predictive factors of infection-related implant removal after ORIF of the extremities. From 2006 to 2017, they evaluated patients treated for upper and lower extremity fractures with the ORIF technique and cases that underwent removal of the implant due to infection after ORIF fixation [8]. A number of predisposing factors to infection-related implant removals have been identified, such as diabetes with chronic complications, liver disease, obesity, anemia deficiency, psychosis, and depression [8]. Diabetes mellitus is a well-known risk factor for complications due to immune dysfunction in the setting of a hyperglycemic environment and impaired wound healing. Liver diseases are another risk factors for post-operative infections [8]. 

In the 67% of cases, the patients were male; it is theorized that this is due to hormonal differences, and it would appear that testosterone has an immune-suppressive effect. Patients who went for removal due to infection had a higher average length of stay than patients who went for removal due to non-infectious causes (17.2 vs. 9.5), and the biggest variation was observed between removals of the tarsals and metatarsals (15.5 days, 27.8 days, vs. 12.3 days); so, in case of infection, we have a higher total charge associated with the episode of care, especially with carpal/metacarpal removals. It is intriguing to note that certain anatomical regions displayed higher rates of implant removal due to infection [8]. The phalanges/hand (5.61%), phalanges/foot (5.08%), and radius/ulna (4.85%) emerged as the locations with the highest incidence of implant removal attributed to infection [8]. Further analysis pinpointed the phalanges/hand (22.0%) and carpal/metacarpal (22.0%) as the most common implant locations associated with infection-related removal, followed by the humerus (11.0%) [8]. These nuanced findings contribute to a comprehensive understanding of the dynamics of infection-related implant removal across diverse anatomical sites. In the realm of implantable devices, the occurrence of infections is a pervasive concern, yet the ramifications of infection following Open Reduction Internal Fixation (ORIF) are especially severe, often culminating in profound morbidity that may necessitate drastic measures, such as amputation, or result in enduring disabilities [8]. The consequences of such infections extend beyond the immediate complications, exerting a lasting impact on an individual’s quality of life [8].

It is noteworthy that despite an overall decrease in implant removal rates due to infection across various fractures, a unique trend emerged in radial/ulnar fractures where removal rates exhibited an increase [8]. This distinctive pattern warrants a more in-depth exploration to understand the underlying factors contributing to this particular divergence in outcomes [8]. The identification of significant predictors for infection-related implant removal sheds light on the multifaceted nature of this issue [8]. Conditions such as diabetes, liver disease, obesity, anemia, depression, and rheumatoid arthritis emerged as noteworthy factors influencing the likelihood of infection-related implant removal [8]. Unraveling the intricate interplay between these health variables and implant-related infections not only enriches our understanding of the clinical landscape but also paves the way for targeted interventions and preventive strategies in high-risk populations [8]. 

## 5. Conclusions

Fracture-related infections are a major complication of trauma surgery. It seems logical to think that timely diagnosis can be critical to case management. For a correct diagnosis, both a careful clinical evaluation to look for signs/symptoms attributable to an infectious condition and instrumental examinations to highlight the site of infection, its extent, and severity are essential. Also deserving of mention are laboratory and microbiological tests that allow identification of the responsible pathogen. The difference between certainty and suggestive criteria is fundamental because treatment will depend on them. Nowadays, the literature has highlighted the role of antibiotic prophylaxis alone and in combination with surgical treatment. Several drug associations have been evaluated. Although the use of antibiotics allows for slowing and controlling the spread of the infectious process, it is necessary to sometimes act more drastically by intervening surgically. In this regard, the DAIR protocol has provided excellent results in the management of this delicate complication.

It is critical to understand, however, that in order to arrive at a diagnosis of certainty, individual parameters are unreliable. Taken individually, they provide little useful information to the surgeon, so for a correct and timely diagnosis, it would be appropriate to evaluate the patient comprehensively from a clinical, instrumental, and laboratory point of view.

## Data Availability

Not applicable.

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
