# Peer review of "Early Management for Fracture-Related Infection: A Literature Review"

_healthcare, 2024, doi:10.3390/healthcare12131306_

Round 1

Reviewer 1 Report

Comments and Suggestions for Authors

Line 24-25 : provide reference please

Line 30-33 : provide reference please

34-35: kindly refine English

43-45 : what is the relation between CDC guideline in treating BJI infection? this guideline is used for prevention 

Kindly all references should be in black color (not red)

Line 54 : give only initials of names, not name of authors

results section should be in materials and methods,  

discussion should be as a literature review section

Discussion:

line 81-87 : not in the scope of review to discuss the prophylactic antibiotic, please ignore it

line 90-92: Usually the use of the single IV AB doesn’t allow to reach suitable therapeutic levels so the use of a combination of IV and local AB with or without a surgical procedure is mandatory :  this is not correct,  surgical treatment is usually mandatory, local antibiotic is debatable,  , authors should discuss retaining materials or not rather than discussing local antibiotic

Line 97:  the organisms cited were in acute infection ? as Coag neg staph is usually in chronic infection , streptcocci in late hematogenous infection , what about pseudomoneus ? 

Line 106-110 : references are not correct

phrase is missing , what do you mean , masquelet technique ? "Literature is quite prolific regarding the use of polymethyl methacrylate bone cement 135 as an efficient tool to prevent infections in open fracture"

Line 140 : phrase is missing, what is the relation of histopathological exam ?  do you mean microbiological examination ? please pay attention that the concensus meeting held in 2018 was also regarding prosthetic joint infection and should not be mixed.

"CRP levels peak on the second day and subsequently revert to baseline after two weeks"  it normalize at 3 weeks

please discuss the rule of alpha definsin and leukoestrase dipstick

Line 193:  i think that the word DAIR is used usually for PJI infection and not for fracture.

211-218 : bacterial organism paragraph is repeated

line 280 : please specify best quinolone (levofloxacine is better than ofloxacine , and cipro for pseudomonseus etc) 

discuss new antibiotic such as dalbavancine, tedizolid, daptomycin 

discuss the poor association of clindamycin and rifampicine (bernard et al)

285:286: add ciprofloxacin as it's the only oral antibiotic can be used

Line 294-318 : this paragraph is not adequete to be placed at the end as should be in risk factor section , also it's repeated

other important points

authors should discussed more regarding surgical management,  the rule of masquelet, ilizarov  in case of resistant infection with bone defect

the rule of vac therapy is crucial to be discussed

new antibiotics such as dalbavancine, daptomycin, tedizolid , 

supressive antibiotic

bone consolidation conisdration and treatment was not detailed 

the title was focused on early managmenet, however, the review focused on risk factors and diagnostic issues,  so i suggest to change title to be more generlized. 

references should be increased for a review article

the discussion is hard to follow and read,  i suggest divide the study into many subheadings (diagnostic, risk factors , antibiotic consideration, treatment etc) 

discuss rule of local antibiotic in a section (including calcium sulfate impergenated with antibiotic, local antibiotic with cements ) 

i'll also consider an illustration of a case of infection that successfully managed 

english should be refined

Comments on the Quality of English Language

it's better to be refined by a native speaker

Reviewer 2 Report

Comments and Suggestions for Authors

The review is well-written with a good methodology. 

Remove the blank page.

I suggest improving the introduction with some more information on the issue of fracture-related infection. 

One main topic is about healthcare-associated infections that in many cases involve patients that undergo orthopaedic surgery. 

These kinds of infections are related also to an increase in costs for treatments and in Italy also for compensations for medical malpractice claims, especially for specific patogens. 

On this last topic I could suggest the following reference. 

Medico-Legal Aspects of Hospital-Acquired Infections: 5-Years of Judgements of the Civil Court of Rome

Healthcare (Switzerland), 2022

Reviewer 3 Report

Comments and Suggestions for Authors

The authors have presented an interesting review of fracture-related infection management. While the information presented appears very worthwhile and helpful, the authors' description of their methods and type of review is greatly lacking. The Methods section needs to be rewritten incorporating relevant information stipulated by the PRISMA 2020 checklist. The title, abstract, and other areas of the manuscript proposal will also require significant revisions. The PRISMA article selection flowchart should also be included. Reference to various "levels of evidence" can be problematic (unless the journal stipulates a certain rating system...), because there are so many different rating schemes. The authors might be better served by either explaining exactly what they mean by "level XYZ" and stipulating which rating scheme they used and how a decision of what level a certain article was was reached. This reviewer looks forward to reading a thoughtfully revised manuscript proposal along the PRISMA 2020 guidelines. 

Comments on the Quality of English Language

While the manuscript proposal is readable, significant English style and grammar editing by an in-field native English-speaker is needed.

Reviewer 4 Report

Comments and Suggestions for Authors

The authors propose to review current literature on fracture-related infections.

The results paragraph is just few lines, no figures, no PRISMA iconography were retrieved (PRISMA flow diagram?). The PRISMA check list should also be added in supplementary materials. I don’t find appropriate to abbreviate antibiotics with AB. I would not use any abbreviation. If the authors decided to unify the results and discussion in a unique paragraph, it should be structured in different paragraphs. The conclusion is too long and inconclusive, what does this review add to the current knowledge? I find it more similar to a narrative review, instead of a systematic review.

43-44 Why sometimes? Guidelines should be applied consistently. Please clarify the meaning.

 “We sometimes use the CDC guideline (US National Institutes for Health Centers for 43 Disease Control and Prevention) to prevent surgical-site infection (SSI), which distin-44 guishes between superficial, deep and organ infections.”  -

54-57 “Three authors collected literature separately (Ladogana T. , Albano F. and Buono C.), and then screened abstracts and full texts. Each paper was evaluated by one independent investigators (Vicenti G.); in cases of 56 disagreement, it was solved by consensus with a third author with extensive expertise in 57 implant-related infections (Solarino G.).”

Writing just the initial of the name and surname would be more appropriate.

59-60 ”Levels II and III of evidence were evaluated for inclusion criteria: reviews, systematic 59 reviews, and meta-analyses focusing on infection in the orthopaedic field “

Studies focusing on infection in the orthopedic field is too generic, it cannot be considered as inclusion criteria. Please explain in detail inclusion and exclusion criteria. Why only reviews and meta-analyses have been included? Is the study a review of reviews? If yes, the key word “review” should be included.

63 “papers with exclusive abstracts available »

Did the authors mean, studies in which only the abstract was retrieved instead of the full text? It should be clarified.

Comments on the Quality of English Language

I would improve the fluency of the manuscript.

Round 2

Reviewer 1 Report

Comments and Suggestions for Authors

authors adressed most issues 

However, i believe this phrase should be corrected "Usually the use of the single IV anti-96 biotic doesn’t allow to reach suitable therapeutic levels so the use of a combination of IV 97 and local antibiotics is mandatory[1]."  as local antibiotic is debatable

i still believe that new antibiotic treatment is vital to be mentioned as a new review article without being "redundant" 

Author Response

The phrase has been corrected

Reviewer 3 Report

Comments and Suggestions for Authors

My concerns from the initial review remain. The manner in which review results are presented does not determine if it is a systematic review or not, only whether it is a meta-analysis. This peer reviewer has planned, conducted, and published protocols and completed manuscripts for systematic reviews where the results were written in a narrative style. Therefore, this reviewer still feels the Methods section should be rewritten following the PRISMA 2020 guidelines. This reviewer looks forward to reading a carefully revised manuscript proposal and congratulates the authors on their hard work. 

Comments on the Quality of English Language

Significant English style and grammar concerns remain, and editing by an in-field native English-speaker is needed. The manuscript proposal is readable, however.

Author Response

As specified in the materials and methods, we have conducted a narrative review. Therefore, all references to PRISMA have been removed.

However, if deemed appropriate, we could create a flow chart to better explain the selection process of the cited works.